# Role of Protein Tyrosine Phosphatase 1B Inhibitor in Early Brain Injury of Subarachnoid Hemorrhage in Mice

**DOI:** 10.3390/brainsci13050816

**Published:** 2023-05-18

**Authors:** Zhong-Hua Zhang, Xiao-Ming Zhou, Xin Zhang

**Affiliations:** 1Department of Neurosurgery, Jinling Hospital, Jinling School of Clinical Medicine, Nanjing Medical University, Nanjing 210000, China; 2Department of Anesthesiology, Jinling Hospital, Jinling School of Clinical Medicine, Nanjing Medical University, Nanjing 210000, China

**Keywords:** protein tyrosine phosphatase 1B, subarachnoid hemorrhage, early brain injury, apoptosis, inflammation factors

## Abstract

Clinically, early brain injury (EBI), which refers to the acute injuries to the whole brain in the phase of the first 72 h following subarachnoid hemorrhage (SAH), is intensely investigated to improve neurological and psychological function. Additionally, it will be meaningful to explore new therapeutic approaches for EBI treatment to improve the prognosis of patients with SAH. To investigate the underlying neuroprotection mechanism in vitro, the Protein tyrosine phosphatase 1B inhibitor (PTP1B-IN-1) was put in primary neurons induced by OxyHb to observe neuroapoptosis, neuroinflammation, and ER stress. Then, one hundred forty male mice were subjected to Experiment two and Experiment three. The mice in the SAH24h + PTP1B-IN-1 group were given an intraperitoneal injection of 5 mg/kg PTP1B-IN-1 30 min before anesthesia. SAH grade, neurological score, brain water content, Western blot, PCR, and Transmission Electron Microscopy (TEM) were performed to observe the underlying neuroprotection mechanism in vivo. Overall, this study suggests that PTP1B-IN-1 could ameliorate neuroapoptosis, neuroinflammation, and ER stress in vitro and in vivo by regulating the IRS-2/AKT signaling pathway, suggesting that PTP1B-IN-1 may be a candidate drug for the treatment of early brain injury after SAH.

## 1. Introduction

Aneurysmal SAH is a form of acute hemorrhagic stroke characterized by a high mortality and morbidity rate and a poor neurologic prognosis [1]. SAH-induced brain injury includes EBI and delayed brain injury. EBI, which refers to acute injuries to the whole brain in the phase of the first 72 h following SAH, could be the primary factor causing the bad prognosis. Many studies have shown that EBI after SAH involves complicated pathologic events including oxidative stress, neuroinflammation, microcirculatory failure, mitochondrial dysfunction, autophagy, and neuronal death [2,3,4]. Due to the complexity of SAH pathogenesis, exploring the specificity and drug targets of SAH experimentally is a difficult task. Additionally, it will be more meaningful to explore new therapeutic approaches for EBI treatment to improve the prognosis of patients with SAH. Since neuroinflammation and neuroapoptosis are the main factors in the pathogenesis of EBI following SAH, brain neuroinflammation reduction and anti-apoptotic therapy are imperative.

PTP1B is a member of the protein tyrosine phosphatase family and has widely attracted attention as a crucial inhibitor of numerous signaling pathways, including insulin, leptin, and tropomyosin receptor kinase B [5,6,7]. Its negative regulatory role on insulin signaling is mainly through dephosphorylation of tyrosine residues on IRS-1/IRS-2 and the major downstream phosphoinositide-3-kinase–protein kinase B(PI3-K)/Akt signal pathway [8]. Moreover, it could dephosphorylate leptin-related receptor kinase Janus kinase 2, the activator of transcription 3 [9]. Recent studies further revealed that PTP1B is highly expressed in the central nervous system and is a regulator of neuroinflammation [10]. Some researchers reported that selective deletion of PTP1B in the neurons of a transgenic mouse model of Alzheimer’s disease ameliorated neuron apoptosis and spatial memory deficits [11]. However, the roles of PTP1B in SAH remain to be studied, and it is quite possible that the inhibition of PTP1B after SAH may play a crucial part in neuroprotective effects by attenuating neuroinflammation and neuroapoptosis.

The ER is the site for protein synthesis and folding, which can guarantee that proteins are appropriately translated [12,13,14]. However, the build-up of unfolded and misfolded proteins during a period of physiological and pathological disturbances will activate ER stress. The activation of ER stress further promotes the UPR, which further reestablishes ER homeostasis [15]. Meanwhile, ER stress is also involved in the pathological progression after SAH [16]. PTP1B has been proven to play a pivotal part in the activation of the UPR, which has emerged as an important relation between insulin resistance and ER dysfunction [12,17,18]. An appropriate UPR is beneficial to normal cell metabolism by eradicating the unfolded or misfolded proteins in the ER. However, heavy ER stress can cause a disproportionate UPR to further induce neuroapoptosis. Both neuroinflammation and ER stress explored in AD was accompanied by the upregulation of the PTP1B [10,19]. However, to the best of our knowledge, the roles of PTP1B in ER stress during SAH are uncertain. Therefore, this experiment aimed to explore the effect of PTP1B inhibition on neuroinflammation, neuroapoptosis, and ER stress after SAH.

## 2. Methods and Materials

### 2.1. Primary Neuronal Culture

Pregnant mice were sacrificed by cervical dislocation. Primary cortical neurons were dissociated from C57/BL6 mice as previously described [20]. Briefly, the cerebral cortex was separated from the brains of embryonic mice (16–18 days). The sample was processed with 0.125% trypsin in a culture vessel for 15 min at room temperature. The combination was filtered through a 22 μm filter and centrifuged at 1000 rpm for 5–10 min. Neurons were distributed onto poly-D-lysine-coated culture vessel at a concentration of 5–7 × 10^5^/mL and suspended evenly in neurobasal media containing 2 mM glutamine (Gibco), B27 supplement (Gibco), and 50 μg/mL penicillin-streptomycin (Gibco). Half of the medium was changed with fresh medium every 3 days. The neurons cultured for 7–9 days were used in in vitro studies. According to our previous study [21], the OxyHb (10 mM) dissolved in the medium was used in the SAH model for 24 h in vitro.

### 2.2. Animals

Male C57BL/6 (22–24 g) mice were obtained from the Animal Experiment Center of Nanjing Medical University. The experimental procedures and animal care were approved by the Ethics Committee of Jinling Hospital. The ethics approval code is 2020JLHGKJDWLS-31. Mice resided in controlled temperature and humidity circumstances (23 ± 2 °C, 40%) with a reversed 12 h/12 h light/dark cycle. Mice were supplied with food and water ad libitum. The mice were accustomed to the situation for at least one week before the experiments.

### 2.3. SAH Model In Vivo

The procedure for the SAH model was as referred in a previous study [22]. Briefly, mice were deeply anesthetized with bromethol (0.5 mL/g, intraperitoneal injection) and placed in a prone position on a stereotaxic frame. We carefully evaluated the state of complete anesthesia until they had no contact reaction to the head, limbs, tail, and whisker when the mice were injected with bromethol. Additionally, the heartbeat and breathing of mice were monitored during manipulation. The model was produced via stereotaxic insertion of a needle at 4.5 mm anterior to bregma in the midline at an angle of 40° in the sagittal plane into the prechiasmatic cistern. Concurrently, another mouse was euthanized as a donor for arterial blood via exposed left ventricular cardiac puncture. The mice were anesthetized again after each group’s modeling time was satisfied. After injecting the mice’s hearts with normal saline for perfusion, we mainly collected the temporal cortex of the hemisphere for ELISA, WB, and PCR research.

### 2.4. Experimental Design and Groups

Our experiments were divided into three separate parts (Experiment 1, Experiment 2, and Experiment 3). First, to observe the expression of PTP1B, p-IRS-2, p-AKT, ER stress, and whether the administration of PTP1B-IN-1 could affect the trend after SAH in vitro, the cells were divided into four groups in Experiment 1: Sham, SAH, SAH + Vehicle (equal amount of DMSO), and SAH + PTP1B-IN-1. Western blot analysis, Double immunofluorescence staining, and PCR were adopted to measure the trend. Second, the expression of PTP1B, p-IRS-2, p-AKT, and ER stress in the temporal cortex after SAH was measured during different timeframes in vivo. Sixty C57BL/6 mice were randomly divided into four groups in Experiment 2: Sham, SAH24h (*n* = 15), SAH48h (*n* = 15), and SAH72h (*n* = 15). Third, the objective was to evaluate the effects of PTP1B-IN-1 added to the SAH24h group through Western blot analysis, Double immunofluorescence staining, PCR, Morris Water Maze, and TEM. Eighty C57BL/6 mice were divided into four groups in Experiment 3: Sham (*n* = 20), SAH24h (*n* = 20), SAH24h + DMSO (equal amount of vehicle) (*n* = 20), and SAH24h + PTP1B-IN-1 (*n* = 20). The mice in the SAH24h + PTP1B-IN-1 group were given an intraperitoneal injection of 5 mg/kg PTP1B-IN-1 30 min before anesthesia. The doses of PTP1B-IN-1 were determined according to the previous research [23].

### 2.5. Cytotoxicity Tests

The mature primary cells (3 × 10^3^ cells/well) were cultured in 96-well plates treated with a PTP1B inhibitor (PTP1B-IN-1) for 24 h. The different concentrations of PTP1B-IN-1 were added to the plate for 24 h, and an equal volume of DMSO was added as a control. To evaluate the cell cytotoxicity, a Cell Counting Kit-8 (Beyotime, Nanjing, China) was adopted following the manufacturer’s instructions. Briefly, 10 μL of the reagent was added to each well, and the culture was incubated at 37 °C for 2 h. Absorbance at 450 nm was evaluated by using a microplate reader (Thermo Fisher, Massachusetts, MA, USA). Cell viability was expressed as a percentage of control (DMSO-treated) cell viability. All experiments were performed in quadruplicate. The PTP1B-IN-1 was dissolved in DMSO to prepare a stock solution at a concentration of 20 mM. Then, the cells were treated with the PTP1B-IN-1 (1 mM/L, 2 mM/L, 4 mM/L, 8 mM/L, 10 mM/L, and 20 mM/L) diluted with DMEM for 24 h.

### 2.6. Neurologic Scoring, Body Weight, SAH Grading Score, and Blood Glucose

Neurological scores were estimated by two researchers blinded to the group using the 18-point modified Garcia scoring system [24]. The Garcia test sets tasks on spontaneous activity, symmetry in the movement of all four limbs, forepaw outstretching, climbing, body proprioception, and response to whisker stimulation (Appendix A). The lower the score, the more serious the neurological function. Bodyweight change was expressed as a ratio of body weight after surgery to body weight before surgery. Then, the SAH grading score was gauged according to Sugawara’s grading scale (Appendix A). Briefly, six parts of the basal brain were scored ranging from 0 to 3, based on the amount of subarachnoid blood clots. Mice with an SAH grade score less than 8 would be excluded from the experiment. Lastly, the blood glucose derived from the left ventricle of mice after being anesthetized was measured promptly using a glucometer (Contour TS, Bayer, Leverkusen, Germany).

### 2.7. Morris Water Maze

To assess cognitive and memory function, animals were trained with a Morris Water Maze (XinRuan Science and Technology Company, Shanghai, China) test with a hidden platform, as referred to in the protocol [25,26]. The platform (10 cm diameter) was 2 cm below the water surface in a circular pool (100 cm diameter) full of warm water (23 ± 1 °C) dyed with white food pigment. During the acquisition phase, the mice were given 60 s to grope in the pool. Once fulfilling the discovery of the platform in less than 60 s, animals could stay on the platform for 15 s. Otherwise, the mice were constrainedly led to the site and stayed there for 30 s. For assessment of the extent of memory consolidation on the 6th day, a probe trial was performed to record and analyze the swimming speed, path tracks, and appearance frequency in the target quadrant.

### 2.8. Brain Water Content

After the mice were deeply anesthetized with bromethol (0.5 mL/g, intraperitoneal injection), the whole brain was separated into the hemisphere, cerebellum, and brain stem and weighed as wet weight. The hemisphere was dried at 110 °C for 3 days and weighed again as dry weight. After that, the water percentage was calculated as: %H_2_O = (wet weight − dry weight)/wet weight × 100%.

### 2.9. Immunofluorescence Staining and TUNEL Staining

After being deeply anesthetized with bromethol, the mice were transcardially perfused with 4% paraformaldehyde. The brains were separated, post-fixed overnight in 4% paraformaldehyde, and dehydrated with 10% and 30% sucrose, respectively, until they sank to the bottom. Coronal sections (10 μm) of the temporal lobe were cut on a microtome (Hitachi, Japan). Meanwhile, the cell slides of primary neurons were immersed in 4% paraformaldehyde.

For immunofluorescence analysis, following treatment with 0.1% Triton X-100 for 5 min, brain sections and cell slides were blocked using 5% BSA for 30–60 min. After being washed with PBS three times for 5 min each, brain sections and cells were subsequently incubated with primary antibody (NeuN 1:100, Santa Cruz Biotechnology) and PTP1B (1:100, Proteintech, catalog number: AB_10642566) overnight at 4 °C. Sections and cell slides were then incubated with secondary antibodies (1:200, Alexa Fluor™ 488, Thermo, Massachusetts, MA, USA A31620; Alexa Fluor™ 594, Thermo, Massachusetts, MA, USA A31632) for 2 h at room temperature, followed by three 5 min washes in PBS. The nuclei were stained with DAPI (1:1000, Sigma, Beijing, China) for 5–10 min. Fluorescence microscopy imaging in the bilateral temporal lobe was obtained under an Olympus inverted microscope system (Olympus, Tokyo, Japan).

As for TUNEL staining, the procedure for processing cell slices was the same as above. According to the instructions of the TUNEL Detection Kit (Beyotime, Nanjing, China), cell slices were incubated with neuron antibody overnight at 4 °C, then with TUNEL for 1 h at 37 °C in the dark. After being washed with PBS three times for 5 min each, they were stained with DAPI for 10 min. The whole process was in the dark. Fluorescence microscopy imaging in the bilateral temporal lobe was obtained under an Olympus inverted microscope system.

### 2.10. Nissl Staining

Brain sections (10 μm) were stained with Cresyl violet solution according to standard instruction and then placed on microscope slides. The morphology and the number of neurons in the bilateral temporal lobe was evaluated under a light microscope. Normal neurons often have round cell bodies and homogeneous nuclei, in contrast to the injured cells. An experimenter blinded to the sample group haphazardly selected four fields (×400). The average number of neurons was determined in four sections.

### 2.11. Western Blot Analysis

After being deeply anesthetized with bromethol (0.5 mL/g, intraperitoneal injection), the mice were transcardially perfused with normal saline. The bilateral temporal tissues were harvested following anesthesia for Western blot, Elisa, PCR, and TEM. The method of whole protein extraction and nuclear protein extraction has been described previously [27]. For each sample, 50 μg proteins were loaded into the polyacrylamide gel and subsequently removed to the PVDF membrane. The membranes were blocked with 5% skim milk for 2 h, then incubated at 4 °C overnight with the following primary antibodies: p-IRS-2 (1:1000 Thermo Fisher, Massachusetts, MA, USA Product: PA5-106094), IRS-2 (1:1000 Proteintech Wuhan, China, catalog number: 20702-1-AP), p-AKT (1:10,000 Proteintech, catalog number: 80455-1-RR), AKT (1:1000 Proteintech, catalog number: 10176-2-AP), CHOP (1:1000 Proteintech, catalog number: 15204-1-AP), GPADH (1:20,000 Proteintech, catalog number: 10494-1-AP), Cleaved Caspase-3 (1:1000, Bioss, Boston, MA, USA lot: BJ03319208), PTP1B (1:2000 Proteintech, catalog number: 11334-1-AP), NF-κB (1:1000 Proteintech, catalog number: 66535-1-Ig), p-NF-κB (1:1000 Thermo Fisher, Product: MA5-15160), β-actin (1:4000 Proteintech, catalog number: 10745-1-AP), Bcl (Affinity, 1:2000, lot#70g9181), Bax (1:2000, Proteintech, catalog number: 50599-2-2g), and Cleaved Caspase-3 (1:1000, Bioss, lot: BJ03319208) overnight at 4 °C. After being washed three times, the membrane was incubated with corresponding horseradish peroxidase-conjugated secondary antibody (goat anti-mouse, 1:5000 Proteintech, catalog number: SA00001-1 and rat anti-rabbit 1:5000 Proteintech, catalog number: SA00001-2) for 1 h at room temperature. The color of bands was measured using ECL Western blotting detection reagents (Thermo Fisher Scientific, Massachusetts, MA, USA). All WB membranes were cut horizontally. Individual gels were run for the separate proteins (p-IRS-2, IRS-2, p-AKT, AKT, p-NF-κB, and NF-κB) because the molecular weight was very close. 

### 2.12. Elisa

The supernate was collected from the bilateral temporal lobe and cell culture medium after being centrifuged at 10,000 rpm for 15 min. The degree of neuroinflammation of IL-1β, IL-6, and TNF-a was evaluated according to the manufacturer’s instructions (Multisciences, Hangzhou, China). Optical density at 450 nm was detected using a microplate reader to analyze the degree of inflammatory factors.

### 2.13. Real-Time PCR

Total RNA extraction was completed with an RNAeasy Animal RNA Isolation Kit with Spin Column (Beyotime, Nanjing, China). The concentration of total RNA was evaluated using NanoDrop (Thermo Fisher Scientific Incorporated, Massachusetts, MA, USA) based on the manufacturer’s protocol. The cDNA synthesis and the RT-PCR analysis were performed using HiScript^®^ III RT SuperMix for qPCR and ChamQ Universal SYBR qPCR Master Mix standard kits (Epizyme Biomedical Technology Co., Shanghai, China), respectively, following the protocols provided by the manufacturer. The cycle threshold (Ct) values were normalized to the GAPDH gene as the loading control. Briefly, 2^−∆∆Ct^ was adopted to calculate the relative expression levels of the gene. The RT-PCR experiment was repeated in quadruplicate. The DNA primer sets (Genscript Biotechnology Co., Ltd., Nanjing, China) used in our experiment are listed in Table 1.

### 2.14. TEM

After being deeply anesthetized with bromethol (0.5 mL/g, intraperitoneal injection), the mice were transcardially perfused with normal saline. Additionally, the brain samples (the bilateral temporal lobe) were instantly detached and immersion fixation was finished at around 1 mm^3^ size through the blade. Samples were immersed in a cold stationary liquid of TEM at room temperature for 2 h. Then, the samples were washed with distilled water before a graded ethanol dehydration series and were immersed in a mixture of half acetone and half resin overnight at 4 °C. After being embedded 24 h later in resin, 70 nm sections were cut and stained with 3% uranyl acetate for 20 min and 0.5% lead citrate for 5 min. Ultrastructure changes of neuronal cells in the temporal cortex were observed under TEM (Hitachi, Tokyo, Japan).

### 2.15. Statistical Analysis

All statistical analyses were performed using GraphPad Prism (version 8.0), ImageJ (version 1.53e), and IBM SPSS (version 25). The dates are expressed as the means ± SD and were analyzed via one-way ANOVA followed by Tukey’s test for multiple comparisons. Statistical significance was inferred at *p* value of <0.05.

## 3. Results

### 3.1. Effect of PTP1B-IN-1 on the Viability of the Cells

A Cell Counting Kit-8 was used to evaluate the effect of PTP1B-IN-1 on the cell viability. The cells were co-cultured with PTP1B-IN-1 at the designated concentrations for 24 h. The viability of the cell was close to the control group when the PTP1B-IN-1 concentration was no more than 2 mmol/L. As the concentration of PTP1B-IN-1 increased, there was a minor decline in cell viability. When the PTP1B-IN-1 concentration was 4 mmol/L, the cell viability was around 80%, signifying that PTP1B-IN-1 was fairly nontoxic to the cells (Appendix A). Therefore, a concentration of 2 mmol/L was adopted in the subsequent vitro experiment.

### 3.2. PTP1B-IN-1 Alleviated Neuroapoptosis, Neuroinflammation, and ER Stress following SAH In Vitro via the IRS-2/AKT Pathway

OxyHb exposure increased the expression of PTP1B (Figure 1A,E,M) compared with that of the Sham group (*p* < 0.05), whereas PTP1B significantly increased the levels of p-NF-кB and decreased the expression of p-IRS-2 and p-AKT (Figure 1H–J) through Western blot and Immunofluorescence staining. Additionally, the PCR results (Figure 1L,M) also revealed the trend of IRS-2 and PTP1B. The IRS-2/AKT insulin signal was inhibited by the dephosphorylation of p-IRS-2 and p-AKT. Additionally, as shown in our previous experiment [28], OxyHb upregulated the levels of PTP1B, p- NF-кB, Bax, Bcl-2, and Cleaved Caspase-3 (Figure 1D–G) as well. By contrast, the knockdown of PTP1B by PTP1B-IN-1 reversed the trend above compared with the SAH + Vehicle group (Figure 1A–J) (*p* < 0.05). Meanwhile, TUNEL (Figure 1K,T) was consistent with the trend of Bax and Cleaved.

The results (Figure 1N–S) also revealed that OxyHb promoted proinflammatory signaling and inflammatory cytokines (IL-1β, IL-6, and TNF-α), whereas PTP1B-IN-1 inhibited OxyHb-induced upregulation in NF-κB and inflammatory cytokines in primary neurons through Western blot and Elisa, compared with that of the SAH + Vehicle group (*p* < 0.05).

As PTP1B resided in the ER membrane, we examined the expression of GRP78 and CHOP, which are the biomarkers of ER stress. The Western blot results (Figure 1A–C) showed that OxyHb speedily upregulated the expression of GRP78 and CHOP. To explore whether PTP1B-IN-1 could decrease OxyHb-induced ER stress following SAH in vitro, we evaluated the expression of ER stress markers once again. By contrast, PTP1B-IN-1 pointedly reduced the OxyHb-induced expression of GRP78 and CHOP (Figure 1A–C) (*p* < 0.05). These results indicated that an upregulation in PTP1B expression caused by OxyHb could trigger heavy ER stress in vitro. Meanwhile, the inhibition of PTP1B greatly attenuated ER stress following SAH associated with OxyHb. 

### 3.3. Mortality Rate and SAH Severity

The overall mortality rate of SAH mice was 14.36%. Three mice were excluded from the experiment due to an SAH score of less than eight in the second part. Four SAH mice were excluded from the experiment due to an SAH score of less than eight in the third part. The SAH grades were not statistically significant in Experiment two (Appendix A) and Experiment three (Appendix A). No significant difference was found in SAH mortality among experimental groups (Table 2). 

### 3.4. Time Course Expression of PTP1B and IRS-2/AKT after SAH In Vivo

The expression of PTP1B in the temporal cerebral cortex was assessed using Western blots and PCR. As revealed in Figure 2A,C,H, there was a noteworthy increase in PTP1B level in the early stage of SAH, which peaked at 24 h after SAH when compared with that of the Sham group (*p* < 0.05). Double immunofluorescence staining revealed that PTP1B was expressed in the temporal cerebral cortex at 24 h after SAH (Figure 2I). Consistently, IRS-2 and AKT in SAH groups were regulated (Figure 2E–G). As shown in Figure 2A–D, the biomarkers of ER stress (GRP78 and CHOP) were upregulated at 24 h, which peaked at 48 h after SAH when compared with those of the Sham group (*p* < 0.05).

### 3.5. PTP1B-IN-1 Alleviated Neuroapoptosis, Neuroinflammation, and ER Stress following SAH In Vivo via the IRS-2/AKT Pathway

To assess the impact of PTP1B-IN-1 on SAH-induced apoptosis in vivo, we first estimated the expression of p-IRS-2, p-AKT, PTP1B, Bax, Bcl-2, and Cleaved Caspase-3 in the temporal cortex. As shown, Western blot analysis (Figure 3F,H,I) showed that PTP1B-IN-1 could pointedly increase the expression of p-IRS-2, p-AKT, and Bcl-2. In addition, the SAH induced the expression of PTP1B (Figure 3E), Bax (Figure 3G), and Cleaved Caspase-3 (Figure 3D), which could be suppressed by PTP1B-IN-1 administration (*p* < 0.05). The trends of IRS-2 and PTP1B were also confirmed using PCR (Figure 3K,L). Meanwhile, TUNEL staining revealed that PTP1B-IN-1 abated the percentage of TUNEL-positive neurons (Figure 3M,T). Double immunofluorescent staining was performed to evaluate the cellular expression of PTP1B with the neuron marker (NeuN) (Figure 3U), astrocyte marker (GFAP) (Appendix A), and Mi/MΦ marker (Iba-1) (Appendix A), respectively. The results of immunofluorescent staining presented that the PTP1B was mostly expressed in neurons at 24 h after SAH. 

Secondly, we evaluated the expression of neuroinflammation and observed that the arterial blood speedily upregulated the expression of p-NF-κB in Western blot (Figure 3A,J). Simultaneously, we measured the changes in IL-1β, TNF-a, and IL-6 contents in the cerebral cortex to assess the effect of PTP1B-IN-1 on neuroinflammation in the 24 h after SAH using Elisa and PCR (Figure 3N–S). The PTP1B inhibitor could decrease inflammatory factor production in the SAH-induced model. NF-κB, which can be regulated by AKT, is a major protein involved in regulating inflammatory responses including IL-1β, TNF-a, and IL-6. In contrast, PTP1B-IN-1 pointedly reversed the trend of neuroinflammation above compared with that of the SAH24 + Vehicle group (*p* < 0.05). These results indicated that an increase in PTP1B expression caused by the arterial blood could trigger dephosphorylation of p-IRS-2 and p-AKT in vivo. Additionally, the dephosphorylation of p-IRS-2 and p-AKT further regulated the downstream signaling pathway of neuroinflammation.

Lastly, to explore whether PTP1B-IN-1 could ameliorate ER stress following SAH in vivo, we estimated the expression of ER stress markers. PTP1B-IN-1 meaningfully decreased the expression of GRP78 and CHOP (Figure 3A–C). These results indicated that the upregulation in PTP1B expression after SAH could the heavy ER stress, which could be reversed by PTP1B-IN-1.

### 3.6. PTP1B-IN-1 Ameliorated Neurological Impairments following 24 h SAH 

The mortality rate of SAH animals did not differ significantly among SAH groups (Table 2). Although we have proved that PTP1B-IN-1 could ameliorate neuronal insulin sensitivity, the underlying mechanism of PTP1B-IN-1 affecting insulin sensitivity was uncertain. Therefore, the influence of PTP1B-IN-1 on neuronal insulin sensitivity required further proof. After treatment of the PTP1B-IN-1 for 24 h, we found prominent changes in PTP1B protein and mRNA levels. Meanwhile, insulin signaling was initiated when insulin bound to the extracellular α-subunit of the insulin receptor, and downstream phosphorylation of IRS-1/IRS-2 at key tyrosine residues was critical for signal transduction. PTP1B played an adverse regulatory role in insulin signaling via dephosphorylation of IRS-2 tyrosine. Though no significant changes in arterial blood glucose and bodyweight change (Figure 4O–P) were observed between the SAH24 + Vehicle group and the PTP1B-IN-1 group, the arterial blood glucose and bodyweight change in the PTP1B-IN-1 group were more stable. Our experiment confirmed that PTP1B-IN-1 could activate insulin sensitivity by upregulating insulin signaling through the activation of the IRS-2/AKT signaling pathway.

To explore the therapeutic effect of PTP1B-IN-1 and its clinical utility, we estimated the neuroprotective properties of PTP1B-IN-1 following 24 h SAH in vivo. Firstly, we tested neurological impairments before mice were sacrificed in each group. Neurological impairments were assessed using a modified Garcia scale, brain water content, and the cognitive and memory function performed by measuring the escape latency and times of crossing the platform. The PTP1B-IN-1 group presented better neurological functions compared with the SAH24h + Vehicle group (Figure 4N). Brain water content was assessed to explore the properties of PTP1B-IN-1 treatment on SAH-induced brain edema lasting 24 h. It was illustrated that brain water content in the SAH24h + Vehicle group was significantly increased compared with the PTP1B-IN-1 group (Figure 4M). Meanwhile, the Nissl staining and the electron micrographs showed the morphology of cells and ultrastructural features following SAH, respectively. Normal neurons often have round cell bodies and homogeneous nuclei, in contrast to the injured cells. Nissl staining (Figure 4B–F) further demonstrated that the outline and the proportion of surviving neurons induced by SAH changed significantly, which could be improved by PTP1B-IN-1. As Figure 4A shows, the electron micrographs showed a pale and watery cytoplasm and swollen organelles, including dilated rough ER fragments, which could be rescued by PTP1B-IN-1. A Morris Water Maze test was performed to evaluate the cognitive and memory function (Figure 4G–J). The probe time was significantly prolonged in the SAH24 + Vehicle group compared with the PTP1B-IN-1 group (Figure 4L). In the probe test, the times of crossing the platform in the PTP1B-IN-1 group were more than those in the SAH24 + Vehicle group (Figure 4K). These data indicate that inhibition of PTP1B meaningfully enhanced the performances of mice in the water maze test following SAH.

## 4. Discussion

Our in vitro experiments and in vivo experiments have presented that targeted regulation of PTP1B could reduce neuronal apoptosis, relieve neuroinflammation and ERS, and thus improve nerve function injury after SAH in mice. Hemoglobin-induced neurotoxicity is driven by intricate, interacting mechanisms that give rise to neuroinflammation, neuroapoptosis, and excessive ER stress in SAH. Meanwhile, neuroinflammation, neuroapoptosis, and ER stress are the main factors in the pathogenesis of EBI following SAH. Additionally, excessive or persistent ERS could further activate a maladaptive reaction and result in neuroapoptosis and neuroinflammation. Therefore, anti-apoptotic therapy and brain neuroinflammation reduction are the imperative treatment for the EBI of SAH. Understanding the mechanisms and pivotal aspects related to SAH is significant to ameliorate prognosis. Additionally, our study was the first to explore the role of PTP1B and the possible mechanisms in the model of SAH in vitro and in vivo. Our novel findings are: (1) PTP1B was expressed in the temporal cortex of mice and peaked at 24 h after SAH. PTP1B played a key role in insulin signaling through the dephosphorylation of tyrosine residues of IRS-2 and AKT, which could deactivate the entire process of insulin signaling; (2) The IRS-2/AKT signaling pathway was the underlying mechanism of the neuroprotection provided by PTP1B-IN-1. PTP1B/IRS-2/Akt activation decreased the phosphorylation of AKT, which was associated with the downregulation of Bcl-2 and the upregulation of inflammatory cytokine levels. PTP1B-IN-1 ameliorated neurological function and attenuated neuroinflammation, neuronal apoptosis, and brain edema at an early stage following SAH. PTP1B could dephosphorylate the phosphotyrosine residues of the activated IR, IRS-2, insulin-like growth factor 1 receptor, colony-stimulating factor 1 receptor, Janus kinase 2, and focal adhesion kinase [29]. Meanwhile, PTP1B played an important role in the regulation of ERS by proinflammatory factors such as TNFα via the NF-κB pathway in SAH. In turn, neuroinflammation could promote the expression of PTP1B in important targets of insulin such as skeletal muscle, the liver, and the hypothalamus through a negative feedback loop [30]. These findings further show that PTP1B plays a crucial part in the crosstalk between ER stress, neuroapoptosis, and neuroinflammation.

PTP1B is encoded by the protein tyrosine phosphatase non-receptor type 1 gene, which creates a 435 amino acid protein with a hydrophobic C-terminal sequence that targets PTP1B to the ER [31]. PTP1B was conventionally considered a routine treatment for diabetes [32] and obesity originally [33]. Recently, it has been considered as a therapeutic objective in other fields including tumors [34], neuroinflammation [10], vascular endothelium protection [35], and neuroprotection. Many researches have revealed that the PTP1B gene is exceedingly expressed in ovarian cancer, gastric cancer, prostate cancer, and breast cancer and is related to poor prognosis [36]. The enzyme has a significant role in regulating numerous cellular processes such as differentiation, inflammation proliferation, apoptosis, and responses to the immune system [37]. However, its function in modulating hemoglobin-induced ERS, neuroapoptosis, and neuroinflammation in SAH remains unexplored. Therefore, we studied the regulatory effect of PTP1B-related signaling pathways in the cerebral cortex and hippocampus, where PTP1B is widespread in neurons [38]. The hippocampus was an important region for learning and remembering spatial signals, which were thought to be the formation of spatial memories. Emerging evidence shows that cellular stresses including ROS, hormonal stress [7], and unfolded or misfolded protein aggregates [11] constitutively stimulate PTP1B, causing neurological function deficits by the IRS. Neuronal PTP1B was upregulated, which accelerated the dysfunction of cognitive and spatial memory in the mice model of Alzheimer’s disease. Recently, some researchers demonstrated that pharmacologic inhibition of PTP1B in microglia effectively reduced detrimental microglial activation, attenuated inflammatory response, and protected neuronal death after cerebral ischemia/reperfusion injury [39]. Our study validated the upregulation of PTP1B in SAH and suggested that neurological function deficits were attenuated by the PTP1B inhibitor. Considering the similar pathogenesis between cerebral hemorrhage and SAH, the PTP1B inhibitor would be a new therapeutic approach and an unaddressed target for the interventions of cerebral hemorrhage. Importantly, there is increasing knowledge about the involvement of other cell types and organs that need a more thorough investigation to be considered for successful interventions [40].

ERS is a significant course for normal physiological function and pathological state, which can be stimulated by temperature stress, ultraviolet exposure, ischemia, and unfolded or misfolded protein accumulation in the ER. Because of the orientation of the PTP1B in the ER membrane, we observed the ER ultrastructure of neurons in the temporal cortex 24 h after SAH using TEM, which was characterized by a rough cytoplasm and swollen organelle, including dilated ER fragments. In contrast, morphological changes would be alleviated by PTP1B-IN-1, showing less edematous ER. Meanwhile, the research highlighted the importance of ERS in the process of insulin resistance and inflammation development, since ER is a major site for the proteins [41]. Mild ER stress could promote cell survival, but excessive ERS could induce the expression of CHOP, which is a biomarker of ERS and can initiate the apoptosis process [42]. Meanwhile, a similar change is induced in the expression of ATF4, which is the upstream gene of CHOP. Our experiment showed that CHOP expression was upregulated and accompanied by neuron apoptosis in the temporal cortex and brain edema at 24 h after SAH, which could be alleviated by PTP1B-IN-1. Previous research concluded that the connection between insulin and ER stress is more intricate than anticipated; the ER stress affected by improving insulin sensitivity is not communicated through the insulin pathway alone. PTP1B was revealed to play a significant part in activating the ER stress response in a complex interaction with insulin resistance in SAH due to its critical location at the surface of the ER membrane. While ERS induced by OxyHb stimulated neuroapoptosis, it also affected the response of neurons to insulin, as evident by the phosphorylation of Akt (Ser473) [35]. However, PTP1B inhibition partially reinstated Akt phosphorylation. In addition, PTP1B inhibition could protect neurons against the pro-apoptotic effects of the OxyHb-induced ERS, as verified by the reduced number of apoptotic cells and blunted increase in the expression levels of vital apoptotic factors. Meanwhile, the obtained results demonstrated an apparent reduction in neuroinflammation, as well as mitigated ERS via the downregulation of Chop upon PTP1B inhibition.

An excessive accumulation of oxidative stress factors and inflammation strongly contribute to insulin resistance development and, in turn, to the neuronal apoptosis [43]. Neuronal apoptosis is a crucial part of many pathological factors, and the degree of neuron survival is closely related to the neurological function [44]. Neuronal apoptosis was considered as the emphasis of research on EBI after SAH. Additionally, apoptosis could affect the permeability of the blood–brain barrier and the degree of brain edema, thereby improving the prognosis of patients. Though Cleaved Caspase-3 was not necessarily a marker of apoptosis after brain injury between day 2 and day 4 after stroke [45], it could be an important marker of apoptosis in the first 24 h after SAH. Additionally, the Bcl-2 family includes antiapoptotic and proapoptotic proteins, which act as indispensable regulators of the intrinsic and mitochondrial pathway of apoptosis. The Bcl-2 family is a multidomain protein that includes BH1, BH2, BH3, and BH4. A variety of BH3 domain peptide analogues are derived from proapoptotic Bcl-2 proteins via the same lipidation approach with their PTP1B binding affinity [46]. Apoptosis is prominent in the brain and is correlated to cognitive and memory dysfunction after SAH. On the one hand, PTP1B-IN-1 could inhibit apoptosis through the IRS-2/AKT pathway. Activated Akt could activate or inactivate its downstream target proteins, including Bad and Bcl-2, via phosphorylation reactivity, thereby regulating cellular signaling. On the other hand, PTP1B-IN-1 could attenuate apoptosis by alleviating ERS by reducing the expression of the majority of CHOP [43]. The experiment in [47] reported that ER stress upregulates expression of PTP1B and impairs glucose uptake, while reference [35] revealed that the PTP1B inhibitor reduced ER stress-induced apoptosis in endothelial cells. For that reason, we can conclude that the PTP1B inhibitor results in increased antiapoptotic gene expression through the upregulation of IRS-2/AKT and the amelioration of ER stress. In a previous study, PTP1B was verified to affect apoptosis via ERK activation [48]. Our results disclosed that treatment with PTP1B-IN-1 could decrease the TUNEL signal, Cleave Caspase-3, and decrease Bax protein expression. We have only explored the intervention that predominantly acts on early brain damage. Our experimental group will continue this experiment about the long-term effects of PTP1B inhibitors on the SAH model.

## 5. Conclusions

PTP1B plays a crucial role in the interaction among endoplasmic reticulum stress, neuronal apoptosis, and neuroinflammation after SAH. Additionally, we generated a graphic (Appendix A) on the molecular pathways and interplay between oxidative stress, PTP1B, and ER changes. Targeted downregulation of PTP1B may be a potential mechanism for alleviating EBI after SAH.

## Figures and Tables

**Figure 1 brainsci-13-00816-f001:**
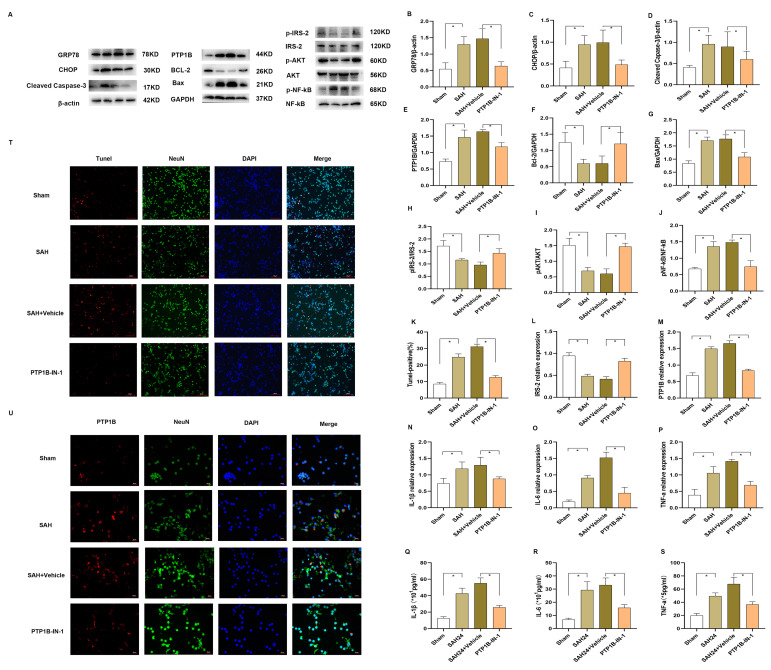
PTP1B-IN-1 alleviated neuroapoptosis and neuroinflammation following SAH in vitro via the IRS-2/AKT pathway. (**A**–**J**) Representative Western blot band and densitometric quantification of the time-dependent expression of p-IRS-2, IRS-2, p-AKT, AKT, p-NF-κB, NF-κB, PTP1B, Bax, Bcl-2, Cleaved Caspase-3, GRP78, and CHOP in primary neurons induced by OxyHb. The expression of PTP1B was upregulated. * *p* < 0.05. Data are presented as mean ± SD, n = 3 per group. Samples derived from the same experiment and the gels were processed in parallel. Quantitative analysis should be normalized to β-actin/GAPDH. Individual gels were run for the separate proteins (p-IRS-2, IRS-2, p-AKT, AKT, p-NF-κB, and NF-κB) when the amount of loading protein was consistent. (**K**) The knockdown of PTP1B by PTP1B-IN-1 reversed the trend of TUNEL compared with the SAH24 + Vehicle group. * *p* < 0.05. Data are presented as mean ± SD, n = 3 per group. (**L**,**M**) The OxyHb affected the expression of PTP1B and IRS-2 through PCR. * *p* < 0.05. Data are presented as mean ± SD, n = 4 per group. (**N**–**S**) The OxyHb promoted proinflammatory signaling and inflammatory cytokines (IL-1β, IL-6, and TNF-α). Meanwhile, PTP1B-IN-1 could obviously inhibit OxyHb-induced upregulation in inflammatory cytokines in primary neurons through PCR and Elisa. * *p* < 0.05. Data are presented as mean ± SD, n = 4 per group. (**T**,**U**) Immunofluorescence revealed that treatment with PTP1B-IN-1 reduced the number of TUNEL-positive cells and the expression of PTP1B in primary neurons, n = 4 per group, scale bar = 50 μm.

**Figure 2 brainsci-13-00816-f002:**
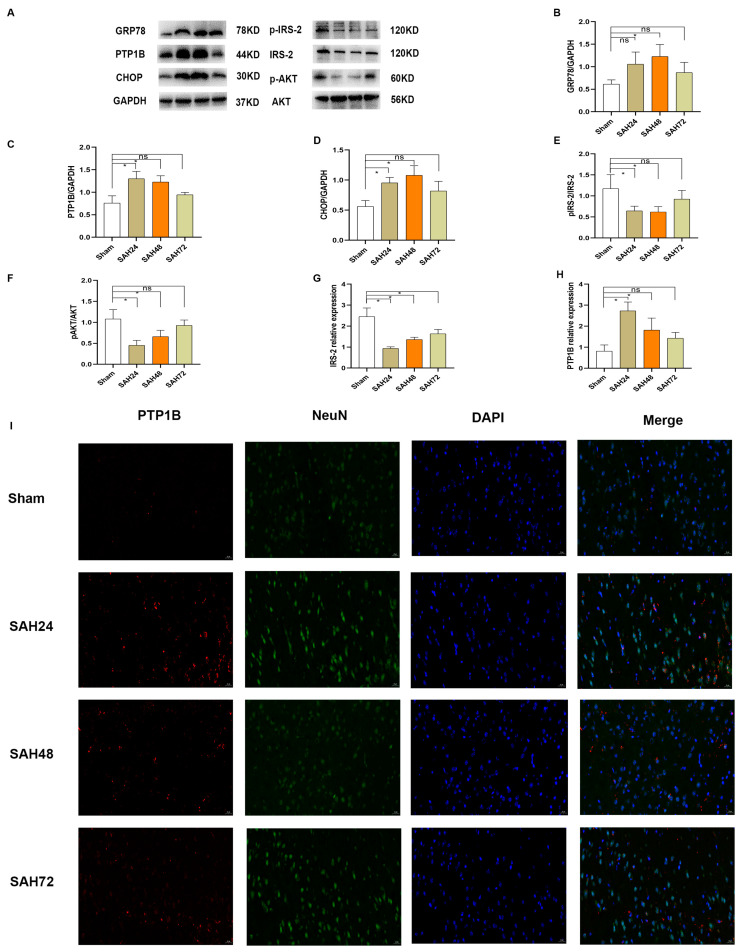
Time course expression of PTP1B and IRS-2/AKT after SAH in vivo. (**A**–**F**) Representative Western blot band and densitometric quantification of the time-dependent expression of p-IRS-2, IRS-2, p-AKT, AKT, and PTP1B after SAH. The expression of PTP1B was upregulated and peaked at 24 h after SAH in different time courses. * *p* < 0.05, ^ns^
*p* > 0.05. Data are presented as mean ± SD, n = 3 per group. Samples derived from the same experiment and the gels were processed in parallel. Quantitative analysis should be normalized to GAPDH. The biomarkers of ER stress (GRP78 and CHOP) were upregulated at 24 h, which peaked at 48 h after SAH. * *p* < 0.05. Data are presented as mean ± SD, n = 3 per group. Samples derived from the same experiment and the gels were processed in parallel. Quantitative analysis should be normalized to GAPDH. (**G**,**H**) The SAH affected the expression of PTP1B and IRS-2 through PCR in different time courses. * *p* < 0.05, ^ns^
*p* > 0.05. Data are presented as mean ± SD, n = 4 per group. (**I**) Immunofluorescence revealed the expression of PTP1B after SAH of different time courses in the bilateral temporal cortex, n = 4 per group, scale bar = 20 μm.

**Figure 3 brainsci-13-00816-f003:**
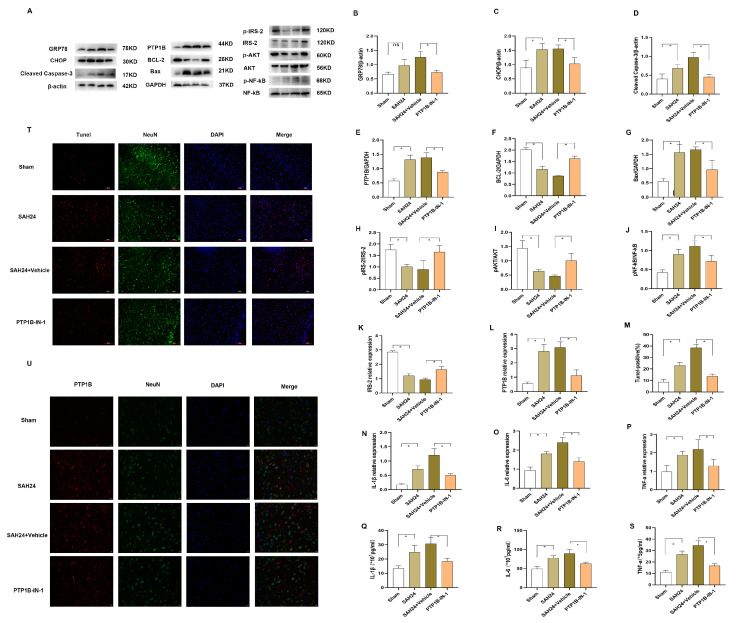
PTP1B-IN-1 alleviated neuroapoptosis and neuroinflammation following SAH in vivo via the IRS-2/AKT pathway. (**A**–**J**) Representative Western blot band and densitometric quantification of the time-dependent expression of p-IRS-2, IRS-2, p-AKT, AKT, p-NF-κB, NF-κB, PTP1B, Bax, Bcl-2, Cleaved Caspase-3, GRP78, and CHOP at 24 h SAH. The expression of PTP1B was upregulated after SAH. * *p* < 0.05, ^ns^
*p* > 0.05. Data are presented as mean ± SD, n = 3 per group. Samples derived from the same experiment and the gels were processed in parallel. Quantitative analysis should be normalized to β-actin/GAPDH. Individual gels were run for the separate proteins (p-IRS-2, IRS-2, p-AKT, AKT, p-NF-κB, and NF-κB) when the amount of loading protein was consistent. (**K**,**L**) SAH affected the expression of PTP1B and IRS-2 through PCR. * *p* < 0.05. Data are presented as mean ± SD, n = 4 per group. (**M**) The knockdown of PTP1B by PTP1B-IN-1 reversed the trend of TUNEL compared with that of the SAH24 + Vehicle group. * *p* < 0.05. Data are presented as mean ± SD, n = 3 per group. (**N**–**S**) The arterial blood promoted inflammatory cytokines (IL-1β, IL-6, and TNF-α) at 24 h SAH in the bilateral temporal cortex through PCR and Elisa. * *p* < 0.05. Data are presented as mean ± SD, n = 4 per group. (**T**) Immunofluorescence revealed that treatment with PTP1B-IN-1 reduced the number of TUNEL-positive cells at 24 h SAH in the bilateral temporal cortex, n = 4 per group, scale bar = 50 μm. (**U**) Immunofluorescence revealed that treatment with PTP1B-IN-1 reduced the expression of PTP1B at 24 h SAH in the bilateral temporal cortex, n = 3 per group, scale bar = 20 μm.

**Figure 4 brainsci-13-00816-f004:**
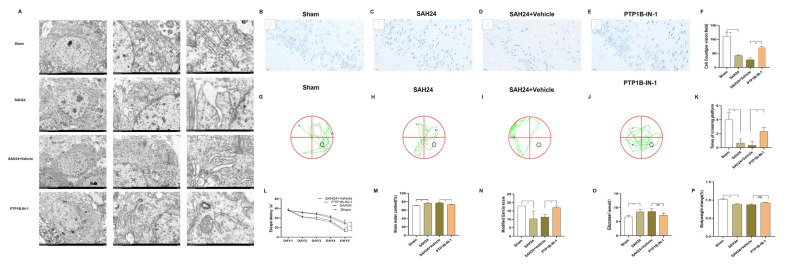
PTP1B-IN-1 ameliorated neurological impairments following 24 h SAH**.** (**A**) The PTP1B inhibitor could improve ER damage after SAH. The TEM (Transmission Electron Microscopy) showed dilated rough ER fragments in the bilateral temporal cortex. PTP1B-IN-1 could alleviate morphological changes, n = 4 per group. (**B**–**F**) The outline and the proportion of surviving neurons induced by SAH changed significantly, which could be improved by PTP1B-IN-1 at 24 h SAH in the bilateral temporal cortex. * *p* < 0.05. Data are presented as mean ± SD, n = 4 per group, scale bar = 20 μm. (**G**–**L**) The arterial blood exposure pointedly prolonged the probe time on the 5th day of the probe test, which could be alleviated by PTP1B-IN-1. PTP1B-IN-1 could obviously shorten the increase in escape latency induced by SAH. The mice in the SAH24 + Vehicle group had a lesser number of entries into the platform in the spatial probe test, which could be improved by PTP1B-IN-1. (**M**–**P**) The PTP1B-IN-1 group presented better neurological functions compared with those of the SAH24h+ Vehicle group. Brain water content, the modified Garcia scale, arterial blood glucose, and bodyweight change could be improved by PTP1B-IN-1. ^ns^ *p* > 0.05, * *p* < 0.05. Data are presented as mean ± SD, n = 3 per group.

**Table 1 brainsci-13-00816-t001:** Real-time PCR primers used in the experiment.

Gene	Forward	Reverse
IRS-2	GCCACAGTCGTGAAAGAGTGA	GTTGGTCGGAAACATGCCAA
PTP1B	CGCCATGGAGATGGAGAAGG	ACACAAGTGTCCTCACCTGG
IL-1β	AAGCTTCCTTGTGCAAGTGT	TAGCCCTCCATTCCTGAAAGC
IL-6	GACAAAGCCAGAGTCCTTCAGA	TGTGACTCCAGCTTATCTCTTGG
TNF-a	GATCGGTCCCCAAAGGGATG	CCACTTGGTGGTTTGTGAGTG
GAPDH	GGGTCCCAGCTTAGGTTCAT	CCCAATACGGCCAAATCCGT

**Table 2 brainsci-13-00816-t002:** Mortality rates in SAH experiments.

Group	Animal Deaths (n)	Mortality Rate (%)
Experiment 2		
Sham	0 (15)	0
SAH24	3 (15)	20%
SAH48	4 (15)	27%
SAH72	4 (15)	27%
Experiment 3		
Sham	0 (20)	0
SAH24	4 (20)	20%
SAH24h + DMSO	6 (20)	30%
SAH24h + PTP1B-IN-1	6 (20)	30%

## Data Availability

All data included in this study are available upon request from the corresponding author.

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
