# Peer review of "Role of Protein Tyrosine Phosphatase 1B Inhibitor in Early Brain Injury of Subarachnoid Hemorrhage in Mice"

_brainsci, 2023, doi:10.3390/brainsci13050816_

Round 1
Reviewer 1 Report
This interesting studies investigates potential neuroprotective effects of protein tyrosine phosphatase 1B inhibition on early brain damage after subarachnoid hemorrhage. A strength of the paper is the broad spectrum of methods being applied and revealing meaningful mechanistic insight. On the other hand, the paper requires major correction as many aspects of the experimental design and methodological details are vague and unclear. This limits interpretation of the results.
Here are my suggestions and concerns.
(1) Please report details on animal numbers (for instance, it is not clear how many animals that survived were excluded due to low Sugawara grading) as well as perioperative pain management procedures. Please also explain the SAH mode l in more detail, it is not clear how exactly this worked and how was appropriate and reproducible “targeting” of the injection ensured. I also doubt that the mouse brain stem is located 4.5 mm anterior of bregma.
(2) Randomization of animals into sham and treatment groups does not make much sense as it automatically prevents allocation concealment. Please also report information about when specific experiments were terminated.
(3) Which cells were used for cytotoxicity tests? Moreover, DMSO can be cytotoxic itself. How was ensured it is an adequate control?
(4) It is unclear what supplement C exactly describes as there is no legend or explanation. What do numbers and arrow heads indicate, what are the six zones used for grading?
(5) Caspase 3 is not necessarily a marker of apoptosis after brain injury (PMID 21256117). This must be mentioned, and potential implications must be discussed.
(6) Why was long-term outcome not investigated? Even if the intervention predominantly acts on early brain damage, a long-term effect should be confirmed.
(7) Parametric tests were used without confirming normal distribution of data. Moreover, most sample sizes (n=3 or 4) are too low to reliably assess normal distribution. Thus, parametric tests must be used. Please report in any reply how individual results changed. Please also provide individual data points (can be done in addition to the “dynamite plunger plots”).
(8) Authors repeatedly talk about “trends”. It is unclear what exactly is meant by that. Please specify. From a statistical point of view, trends do not exist unless being properly defined (e.g., 0.05 </= p < 0.1)
(9) Please provide full (uncropped) Western blots as a supplement.
Minor aspects
(10) Please report methods of animal culling for gaining primary cortical neurons for the in vitro experiments.
(11) Why did animal experiments adhere to “Good Clinical Practice and the Declaration of Helsinki” (lines 98-99)? This does not make sense!
(12) What does “After the perfusion…” (line 110) refer to / mean?
(13) It would be great if Supplements and Figures (or alternatively, the text) could be arranged in a way allowing to mention all panels in chronological manner.
(14) Most scale bars are far too tiny to be identified. The graphs and TEM micrographs in Figure 4 are also very small and no details can be identified. Many graphs and images seem distorted.
(16) Authors may also wish to briefly discuss the relevance of their findings for other forms brain hemorrhage such as ICH (PMID 35158309).
The language of the paper requires massive editing as it is peppered with mistakes, clumsy or unclear phrases, and grammatical errors. For instance:
(a) “Therefore, it will be more meaningful….” (line 17): “Therefore” does not logically connect to the previous sentence and it is not clear compared to what the suggested exploration will be “more meaningful”. Please rephrase.
(b) “Experiment 2 and Experiment 3” (line 21): completely unclear what this means – this is an abstract!
(c) “… 3. the mice in…” (line 21): case sensitivity, should read “The mice…”
(d) “in vitro” (line 24): probably meant is “in vivo”
(e) “abundant” (line 40): inappropriate term, should better read “numerous” etc.
(f) “…is an experimentally impractical and thought-provoking mission as a reason for the complexity of pathogenesis.” (lines 44-45): meaning is unclear
(g) “Meanwhile, it could…” (line 56): shall better read, “Moreover, PTP1B dephosphorylates…”
(h) “In the meantime, …” (line 59): we do not know whether that happened in the meantime, and it is also irrelevant. The phrase can be omitted.
Please note these are just some examples collected from the first two pages. It is impossible to list all errors here. It is recommended that a professional language correction service with experience in medical writing is consulted.
Reviewer 2 Report
The authors have presented tyrosine phosphatase 1B inhibitor role in early brain injury of subarachnoid hemorrhage and makes for an interesting read.
Author Response
We sincerely appreciate the valuable comments. Thank you very much!
Reviewer 3 Report
The authors present their findings on oxyhemoglobin induced injury of neuronal cells in a mouse model of SAH to simulate EBI after aneurysmal hemorrhage and rescue with PTP1B inhibitor. Overall these are very impressive findings and multi-staged supporting experiments. We raise the following questions for the authors:
1. How do the authors know 24 hours of oxyHB is sufficient to simulate SAH conditions. Why not deoxyHB or a combination, why not 12 hours or another time point.
2. Could the authors generate a graphic on the molecular pathways and interplay between oxidative stress, PTP1B, and ER changes. This will help readers understand these experiments better.
3. How do the authors ensure that the degree of SAH is similar in each mouse. We understand the injection method but do the authors have proof it provides consistent results. If SAH is different from mouse to mouse, it causes skepticism to these findings. Also why is the temporal lobe used for cellular testing...why not the cerebellum, or frontal lobes.
Overall well presented. Minor flaws such as below. Not sure what this sentence means
Exploring a specific and druggable target for SAH is an experimentally impractical and thought-provoking mission as a reason for the complexity of pathogenesis.
